# Effect of Relative Humidity on Mechanical Degradation of Medium Mn Steels

**DOI:** 10.3390/ma13061304

**Published:** 2020-03-13

**Authors:** Qingyang Liu, Juanping Xu, Liancheng Shen, Qingjun Zhou, Yanjing Su, Lijie Qiao, Yu Yan

**Affiliations:** 1Beijing Advanced Innovation Center for Materials Genome Engineering, Institute for Advanced Materials and Technology, University of Science and Technology, Beijing 100083, China; liuqingyang567@163.com (Q.L.); 18811390215@163.com (J.X.);; 2Division of Technological Sciences, Bureau of Frontier Sciences and Education, Chinese Academy of Sciences (CAS), Beijing 100864, China; 3Baoshan Iron & Steel Co., Ltd., Research Institute, Shanghai 201900, China

**Keywords:** medium Mn steel, relative humidity, fracture morphology

## Abstract

Medium Mn steels have been considered as the next-generation materials for use in the automotive industry due to their excellent strength and ductility balance. To reduce the total weight and improve the safety of vehicles, medium Mn steels look forward to a highly promising future. However, hydrogen-induced delayed cracking is a concern for the use of high strength steels. This work is focused on the service characteristics of two kinds of medium Mn steels under different relative humidity conditions (40%, 60%, 80% and 100%). Under normal relative humidity (about 40%) at 25 °C, the hydrogen concentration in steel is 0.4 ppm. When exposed to higher relative humidity, the hydrogen concentration in steel increases slowly and reaches a stable value, about 0.8 ppm. In slow strain rate tensile tests under different relative humidity conditions, the tensile strength changed, the hydrogen concentration increased and the elongation decreased as well, thereby increasing the hydrogen embrittlement sensitivity. In other words, the smaller the tensile rate applied, the greater the hydrogen embrittlement sensitivity. In constant load tests under different relative humidity conditions, the threshold value of the delayed cracking of M7B (‘M’ referring to Mn, ‘7’ meaning the content of Mn, ‘B’ denoting batch annealing) steel maintains a steady value of 0.82 σ_b_ (tensile strength). The threshold value of the delayed cracking of M10B significantly changed along with relative humidity. When relative humidity increased from 60% to 80%, the threshold dropped sharply from 0.63 σ_b_ to 0.52 σ_b_. We define 80% relative humidity as the ‘threshold humidity’ for M10B.

## 1. Introduction

With the rapid development of vehicles, the lightening of weight, energy saving and safety improvement are urgently required in the automobile industry. Over recent decades, automobile manufacturers have mainly considered the cost, formability and corrosion resistance of advanced high strength steels (AHSSs) with a good combination of tensile strength and ductility. In general, AHSSs are classified into categories of first-generation AHSSs, second-generation AHSSs and third-generation AHSSs (3rd-G AHSSs) [1]. Of the 3rd-G AHSS steels, medium Mn steel (MMS), which, as far as we know, was first reported by Miller [2], has received much attention because it has dual phase consisting of α ferrite and retained austenite, γ_R_. To date, several researchers have investigated the effects of initial martensite microstructure [3], intercritical annealing conditions such as temperature [4,5,6], time [3,7,8,9,10,11], heating rate [12] and cooling rate [13], and alloying elements such as Mn, C [13], Al [14,15], and Si [16], on the microstructure and tensile properties of MMSs.

Another phenomenon for this type of material is the strength decline due to the presence of hydrogen [17]. Dutton reported that hydrogen is absorbed by metals either during fabrication or service and often causes delayed failure after prolonged periods at stresses which are well within normal load bearing capabilities [17]. The initiation of localized cracking in hydrogenated high strength steel was dependent on the development of a critical hydrogen concentration. So the stress was believed to influence the delayed failure process by providing the means for grouping the hydrogen [18]. Chu [19] agreed with the above-mentioned points and concluded that the hydrogen-induced reduction of yield stress and delayed cracking is dependent on the strain rate, load stress and test temperature, i.e., it is controlled by hydrogen diffusion.

As for corrosion behaviour in the service environment, many researches has studied the effect of dry–wet cycle, full immersion and simulated coastal-industrial atmosphere for different kinds of steels, such as carbon steel [20], and MnCuP weathering steel [21]. A corrosion kinetic model was also discussed by Cai et al. [22] in relation to atmospheric corrosion in dynamic environments and the result demonstrated that the relative humidity was the most influential factors on corrosion. In a real service condition, hydrogen is produced from the decomposition of water, which is closely related to the relative humidity in the environment. Relative humidity (RH) is the ratio of actual water vapor to saturation water vapor pressure at a given temperature. It is especially variable between seasons and at different times of day, with pronounced local variations, particularly in coastal and mountain regions. Hydrogen entry into steels has been studied to evaluate the frequency of delayed fracture under actual atmospheric environments [23]. Site and time dependence of hydrogen absorption [24] and difference in absorbed hydrogen concentrations at various portions in actual bolts exposed at a seashore were reported [25].

RH values below 90% and above 20% are to be expected in almost any part of the world. In particular, early morning humidity of 80 to 100% is common in most coastal and low-lying areas. But only a few studies have focused on the atmospheric corrosion of MMSs and the effect of RH on service performance. In this work, we will systematically investigate the influence of hydrogen from different RH conditions on the hydrogen-induced delayed cracking of MMSs.

## 2. Experimental

### 2.1. Materials

MMS thin plate were supplied by Baoshan Iron&Steel Co., Ltd. In this study, the effect of the RH on hydrogen-induced delayed cracking of MMSs was investigated. The main differences between the two kinds of Mn steels are the Mn and V content. From the microstructure observation by our previous work [26], M7B share the same microstructures with M10B (‘M’ referring to Mn, ‘10’ meaning the content of Mn, ‘B’ denoting batch annealing), which is uniformly distributed with globular-shaped ferrite and retained austenite, with a grain size of about 1–3 μm. Table 1 listed the chemical composition and mechanical properties of the materials. The samples prepared under these two annealing conditions are designated as “M7B”, and “M10B”. 

The heat treatment involved hot-rolled, intercritically batch-annealed, and cold-rolled processes, as follows. The slabs, after steelmaking, were heated to 1473 K for 2 h and hot-rolled at a temperature of 1153 K. The intercritically batch annealing was conducted at 893 K for 12 h. The thickness for hot-rolled sheets was 2.8 mm, consisting of austenite and ferrite. Then the sheets were pickled and cold-rolled to a thickness of 1.4 mm. The cold-rolled MMS sheets were heated to 893 K and held for 12 h. After heat treatment by manufactures, we received the thin plate medium Mn steels with a dual phase of austenite and ferrite [26]. 

The tensile test was prepared to obtain the mechanical properties and stress–strain curves of MMSs. X-ray diffraction (XRD: UItime IV, RIGAKU, Tokyo, Japan) was performed to obtain the volume fraction of retained austenite. A study on the hydrogen permeability of specimens was conducted to obtain the effective hydrogen diffusion coefficient [27]. The sample shape was designed like a coin with a diameter of 20 mm and the thickness of samples for M7B and M10B is 1.28 mm and 0.44 mm, respectively. Prior to the permeation test, the specimen was ground with SiC paper up to 5000 grits, then ultrasonically cleaned in acetone. In order to avoid the corrosion or passivation of the sample in the solution, nickel coating is usually required on the surface of the sample. The nickel electroplating solution is 250 g NiSO_4_-7H_2_O + 45 g NiCl_2_-6H_2_O + 40 g H_3_BO_4_/L, with a current density of 10 mA/cm^2^ for 60 s. The hydrogen-charging solution on the charging side was 0.2 molL^−1^ NaOH + 0.22 gL^−1^ thiourea (H_2_NCSNH_2_) with 5 mAcm^−2^ charging current density at room temperature, approximately 25 °C, while the solution in the escaping side was 0.2 mol L^−1^ NaOH [28].

### 2.2. Exposing Test in Humidity Condition

Saturated aqueous solution of inorganic salts offers convenient secondary standards for the calibration of instruments for the measurement of RH [29]. Saturated salt solution can provide a constant RH value for enclosed space at a certain temperature [30,31]. Saturated sodium bromide solution (NaBr) was chosen to be kept a constant RH of 60% across a wide range of temperature (5–80 °C), saturated potassium chloride (KCl) solution was kept at 80% RH and saturated potassium sulfate (K_2_SO_4_) solution was kept at 100% RH [24]. 

Samples 1.3 mm thick (M7B, for example) were exposed to different humidity conditions to observe the content of hydrogen entry. Samples were taken out to measure hydrogen content (diffusible hydrogen in wppm) after different exposing times: 0 h, 6 h, 12 h, 24 h, 36 h, 48 h, 60 h and 72 h. Due to the dispersion of experimental data for the hydrogen test, the average value of minimum three samples was used and scale bar was shown in Section 3.2. Temperature has a great influence on the adsorption and diffusion behaviour of hydrogen. In order to reduce the interference of temperature, the experiment was carried out at a constant temperature for 25 °C. Hydrogen concentration was measured using the G4 Phoenix DH (Bruker, Stuttgart, Germany) diffusion hydrogen analyzer. The G4 hydrogen detector collects hydrogen from the sample by means of an overheating extraction method. The sample is heated by infrared heating, and the hydrogen inside the specimen escapes selectively according to the temperature. The gas carried by the carrying nitrogen is detected by the thermal conductivity detector. Diffusible hydrogen concentration was measured by G4 at 400 °C with a heating rate of 50 K s^−1^, and the measurement time is 20 min. 

### 2.3. Constant Load Test and Slow Strain Rate Tensile Test 

Constant load test (CLT) is a reliable method to obtain the threshold stress and evaluate the sensitivity of hydrogen-induced delayed cracking. Considering the experimental cycle and accuracy, the cut-time was set to no more than 200 h. The slow strain rate tensile test (SSRTT) is a classical method for exploring susceptibility to hydrogen embrittlement, which was determined by measuring the fracture strength drop, the reduction of the area and the elongation of the specimen under different humidity conditions in this study. The tensile rate was 10^−4^ s^−1^ in air, 10^−5^ s^−1^ and 10^−6^ s^−1^ for M7B, 10^−4^ s^−1^ and 10^−6^ s^−1^ for M10B in wet air. Two parallel samples are used in CLT and SSRTT and the results showed great consistency. To ensure the data accuracy and repeatability, three specimens were prepared for hydrogen concentration after test. The specimen was about 1cm long, which was cut near the fracture. The prepared specimens were immediately washed with acetone and absolute ethanol, and then were blown dry with cold air [32]. The above process was completed in 10 min.

To facilitate clamping, a circular hole was machined on both sides of the sample, whose specific dimensions are designed according to ASTM E8. The gauge part of the sample was 20 mm long, 5 mm wide and approximately 1.5 mm thick. The schematic diagram of MMSs tensile specimen can be found in our previous work [26]. Their surfaces were wet-grounded up to 2000 grits with SiC paper, then ultrasonically cleaned in acetone and ethanol. The tensile direction was parallel to the rolling direction. The specimens prepared for SSRTT and the CLT were exposed to different RH environments for 72 h before testing was carried out. And both tests were conducted still under humid environments. The fracture morphology obtained after SSRTT and CLT were observed by the field emission scanning electron microscopy (FE-SEM; ZEISS DSM) at working voltage of 20 kV and working distance of 12–13 mm [32]. 

## 3. Results Discussion

### 3.1. Mechanical Properties

According to an early study, M7B and M10B are uniformly distributed with globular-shaped ferrite and retained austenite, with a grain size of about 1–3 μm. They consisted of a dual phase of austenite and ferrite, similar to other MMSs. XRD can be used to measure the volume fraction of retained austenite in MMSs. According to Figure 1, the volume fraction of austenite for M7B was 24.2%, and 45.2% for M10B. 

The nominal stress–strain curves of MMSs are shown in Figure 2. Mn, as an austenitic stabilization element, has a direct influence on the volume fraction of retained austenite and its stability. The higher content of Mn ensures the higher content of austenite. More austenite was deformed during tensile test, resulting in greater elongation. Thus, M10B exhibited greater elongation than M7B. 

The different content of austenite contributed to effective different hydrogen diffusion coefficient for the MMSs. According to the result of hydrogen permeation test, as shown in Figure 3, we can see from the curve that the anode current rises along with the hydrogen-charging process until a steady state is reached. t_0.63_ was the corresponding time of Ia/I∞ = 0.63 using the Equation (1) [27,33,34,35]. It is worth noting that Equation (1) is for the case of a uniform solution without trapping according to the McNabb–Foster equation [35]. The effective hydrogen diffusion coefficients of M7B and M10B were 1.08 × 10^−7^ cm^2^/s and 4.41 × 10^−9^ cm^2^/s, respectively.
(1)$$D=\frac{L^{2}}{6t_{0.63}}$$

### 3.2. Exposure Test Results

After exposing to different humidity environments, samples were taken out to analyze their hydrogen content, which enters the material during the exposure process. Figure 4 shows the curve of diffusible hydrogen content (wppm) as a function of exposure time. Under different RH conditions, the hydrogen content in M7B steel varies with time. At 40% RH, hydrogen concentration changes slowly with time, while, at 60% RH and 80% RH, hydrogen content increases rapidly with time. At 100% RH, hydrogen concentration changes with the fastest growth rate and reaches saturation in a relatively fast time (24 h). In general, the hydrogen content in the sample increases gradually with time. Under different RH conditions, the hydrogen concentration changes rapidly in the early stages and reaches a steady-state value in about 60 h. In the case of a higher RH value than 40%, the stable value of the hydrogen content is about 0.8 ppm; and, in case of low RH of 40%, the hydrogen density is about 0.4 ppm. The hydrogen concentration is far from the saturated hydrogen concentration in M7B. Therefore, in a more moderate environment in terms of the influence of humidity, the actual entry of hydrogen is not too big; and, in the presence of external load conditions, the occurrence of stress-induced hydrogen diffusion results in the significant increased entry of hydrogen into the sample.

### 3.3. Slow Strain Rate Tensile Test 

#### 3.3.1. Slow Strain Rate Tensile Test Results under Different Humidity Conditions for M7B

Figure 5 shows the stress–strain curves of M7B at 60%, 80% and 100% humidity and at different strain rates. The data for hydrogen embrittlement sensitivity, tensile strength, elongation and hydrogen concentration are shown in Table 2. According to the experiment, the fracture strength of M7B steel does not change significantly with an increase in RH, but the elongation decreases accordingly. The hydrogen embrittlement sensitivity also increases gradually due to the increase of hydrogen concentration with the increase of RH. While the sensitivity of hydrogen embrittlement in this study is obtained by the loss of ductility, which can be seen as one kind of Embrittlement Index and can be calculated by the following Equations (2) and (3) [26]:(2)$$\delta =\frac{l_{0}-l^{\prime }}{l_{0}}$$
(3)$$loss\;of\;ductility=\frac{\delta_{0}-\delta_{H}}{\delta_{0}}\times 100\%$$
where l_0_: the length of gauge part before SSRT; l’: the length of the gauge part after SSRT; δ: the elongation of the specimen; δ_0_: the elongation of the specimen in air; δ_H_: the elongation of the specimen under different relative humidity condition.

When the strain rate decreases from 10^−5^ s^−1^ to 10^−6^ s^−1^, the material elongation decreases more obviously and the hydrogen embrittlement sensitivity increases significantly. The change of RH and strain rate have little effect on the fracture strength of M7B, but significant effect on the elongation and hydrogen embrittlement sensitivity. It can be seen that the increase in RH and the decrease in strain rate will lead to an increase of hydrogen concentration and hydrogen embrittlement sensitivity of the material, thus reducing its service performance.

By comparing the tensile fracture of M7B under different conditions, SEM analysis was carried out on the tensile fracture in air and on the tensile specimen with a slow strain rate under different humidity conditions, shown in Figure 6.

According to Figure 6a,b, for M7B steel, there is necking at the fracture under air, which presents dimples; these are typical of ductile cracking. When M7B steel is in service under humidity conditions, the macroscopic fracture reveals a middle arc shape similar to the “sole” shape, shown in Figure 6c,e,g,k, in which the edge is a brittle fracture zone with transgranular fracture (cleavage/quasi-cleavage shown in Figure 6f,j,l and intergranular fracture characteristics (shown in Figure 6j,l), while the center is a ductile fracture caused by rapid cracking (shown in Figure 6h,i). 

Cleavage/quasi-cleavage fracture is classified as a brittle transgranular fracture by separation across well-defined crystallographic planes, which occurs under high stress and related with dislocations [36]. Quasi-cleavage fracture surface is produced by a cleavage crack propagating through the complex microstructure (as shown in Figure 6l) where the perfect cleavage facets cannot be formed [37]. Intergranular fracture along prior-austenite grain boundaries is the most common fracture mode for HE of high-strength steels [38]. It is considered that the presence of absorbed hydrogen weakens grain boundaries [39], featured with clear trail of grain fall-off (shown in Figure 6l) or segregation of participates at the grain boundaries [40] to form the continuous weak hydrogen bonding region. 

The area of brittle zone tends to increase with the increase of RH, which accelerated the entry of hydrogen into the sample by hydrogen adsorption and absorption. The detailed process is shown in Figure 7a the reactions of H_2_ + M → H2M, H2M + M → 2HadM, HadM → MHab, MH_ab_→M+H, where H_ad_/H_ab_ mean adsorbed/absorbed hydrogen.

The fracture surface shows fully ductile mode characterized with dimples in Figure 6b. With the increase of RH and decrease of strain rate, the fracture morphology exhibited brittle mode characterized with cleavage/quasi-cleavage and intergranular fracture. When hydrogen atoms entered the samples, the fracture initiation formed near the surface of the material, which lead to brittle cracking. Due to the increase in RH, the hydrogen concentration at the tip of crack increases, and the influence of hydrogen on the sample fracture model can be obtained, as shown in Figure 8.

Akiyama et al. suggested SSRT under a wet condition using specimens with corrosion products formed under actual atmospheric environments or cyclic corrosion tests, to reproduce enhanced hydrogen entry influenced by the corrosion layer and to homogenize the hydrogen distribution in a specimen [25]. Hydrogen entry is influenced by relative humidity, corrosion rate and pH of the inner dust layer, as well depended upon time, season and exposure site [25]. Higher humidity for hydrogen homogenization process results in higher diffusible hydrogen concentration. As shown in Figure 7b, when hydrogen entry proceeds due to corrosion, the surface hydrogen concentration becomes high and when the hydrogen entry is suppressed the surface hydrogen concentration seem to be decreased than that in the interior because of diffusion mainly controlling hydrogen release [41].

#### 3.3.2. SSRTT Result under Different Humidity for M10B

Figure 9 shows the stress–strain curves of M10B under three RH conditions and two strain rate conditions (10^−4^ s^−1^ and 10^−6^ s^−1^), while Table 3 organizes the data on the mechanical properties, hydrogen concentration and hydrogen embrittlement sensitivity. According to the experimental results at a strain rate of 10^−4^ s^−1^, with the increase of RH, the fracture strength of M10B steel reduced significantly. Given that the decrease of elongation causes the earlier fracture during the plastic deforamtion at a strain rate of 10^−6^ s^−1^, the fracture strength changes little but the hydrogen concentration and hydorgen embrittlement sensitivity gradually increases with the increase of RH. When the strain rate decreases from 10^−4^ s^−1^ to 10^−6^ s^−1^, the material elongation decreases more obviously and the hydrogen embrittlement sensitivity increases significantly. The increase of RH and decrease of strain rate accelerate the hydrogen degradation of material by reducing the elongation and increasing the hydrogen embrittlement sensitivity.

By comparing the tensile fracture of M10B under different conditions, SEM analysis was carried out on the tensile fracture in air and on the tensile specimen with a slow strain rate under different humidity conditions; the fracture surface of M10B was shown in Figure 10.

As shown in Figure 10a,b, the fracture morphology of M10B presents relatively uniform dimple characteristics in air. However, from a macro perspective in Figure 10c,e,g,i the fracture has more tearing layers and a delamination fracture pattern, which remains at a low strain rate or occurs after the introduction of humidity. Some researchers reported that the delamination fracture appeared at half thickness of the cross section, where stress exceeded yield strength to accelerate the occurrence of the crack source in the half thickness of the cross section of the specimen during tension test [42]. The fracture surfaces of the samples after test in higher RH environment are severely corroded and show different fracture modes from that in air. The fracture surfaces showed transgranular (cleavage/quasi-cleavage) and intergranular features in 60% RH at a strain rate of 10^−4^ s^−1^, shown in Figure 10d. While the RH increases from 60% to 100% or the strain rate decrease from 10^−4^ s^−1^ to 10^−6^ s^−1^ at 60% RH, the fracture surface of M10B shows a total brittle mode characterized by transgranular (cleavage/quasi-cleavage) and intergranular fracture in Figure 10f,h,j.

### 3.4. Constant Load Test (CLT) 

#### Constant Load Test Result under Different Humidity for M7B and M10B

For the CLT, the test protocol followed was as discussed above. The results of the CLT test for M7B and M10B under three RH conditions are recorded and the relationship between applied stress σ/σ_b_ (the ratio of applied stress to tensile strength) and fracture time t_F_ is shown in Figure 11. The figure is a fit curve creating using Origin software.

The corresponding stress threshold values under different RH conditions were obtained. Under the conditions of 60% RH, 80% RH and 100% RH, the threshold stress factors were 0.84, 0.8 and 0.79, respectively. It can be seen that for M7B steel, there was no obvious change in the threshold stress value, along with the increase in RH. In a high humidity-containing service environment for M7B, the threshold stress does not significantly decrease with an increase in RH, and the change in hydrogen content is not obvious; thus, M7B has a good delayed cracking resistance.

Similarly, the corresponding stress threshold values of M10B under different RH conditions were obtained. Under the conditions of 60% RH, 80% RH and 100% RH, the threshold stress factors were 0.63, 0.53 and 0.52, respectively. Under the same humidity environment, the threshold stress factor of M10B is less than that of M7B. It can be seen that, for M10B steel, the change in the threshold stress value decreased significantly with an increase in RH. At 80% RH, the threshold stress value decreases to a large extent, while hydrogen concentration increases synchronously, which can be maintained in a higher humidity range. 80% RH can be taken as the critical humidity of M10B. The service performance of materials in the environment fluctuates greatly at this level of humidity.

## 4. Conclusions

In this paper, the environmental corrosion sensitivity and hydrogen-induced delayed cracking behavior of M7B and M10B steels were studied by a combination of characterization methods, including X-Ray Diffraction analysis, the hydrogen permeation test, the humidity exposure test, the SSRTT and the CLT under humidity conditions. The conclusions are as follows:Different Mn contents will cause changes in the volume fraction of austenite. The content of austenite in M10B steel is much higher than that in M7B. The elongation of M10B is higher than M7B, because more austenite tends to deform during the tensile test. The effective hydrogen diffusion coefficients of M7B and M10B steel are 1.08 × 10^−7^ cm^2^/s and 4.41 × 10^−9^ cm^2^/s, respectively. The behavior of hydrogen diffusion is also influenced by different content of austenite, which hinders the diffusion of hydrogen.When exposed to air, the hydrogen concentration in the M7B sample is 0.4 ppm. When M7B is exposed to 60% or higher RH, the hydrogen concentration increased to 0.8 ppm. For M7B, an increase of RH and a decrease of strain rate accelerate the hydrogen degradation with an increase of hydrogen concentration of the material, thus reducing its service performance.Similar hydrogen degradation was also found in M10B, as fracture strength and elongation decrease gradually under strain rate of 10^−4^ s^−1^ with the increase of relative humidity and hydrogen concentration, resulting in final failure. Elongation decreases gradually under a strain rate of 10^−6^ s^−1^, with higher relative humidity increasing hydrogen content in M10B.Due to the entry of hydrogen into samples in environments with high relative humidity, fracture morphology changes from ductile mode with obvious dimples to brittle mode characterization with intergranular fracture.For M7B, the threshold stress does not significantly decrease with an increase in RH, but shows a better delayed cracking resistance than M10B. Eighty percent RH can be accepted as the critical humidity for M10B.

## Figures and Tables

**Figure 1 materials-13-01304-f001:**
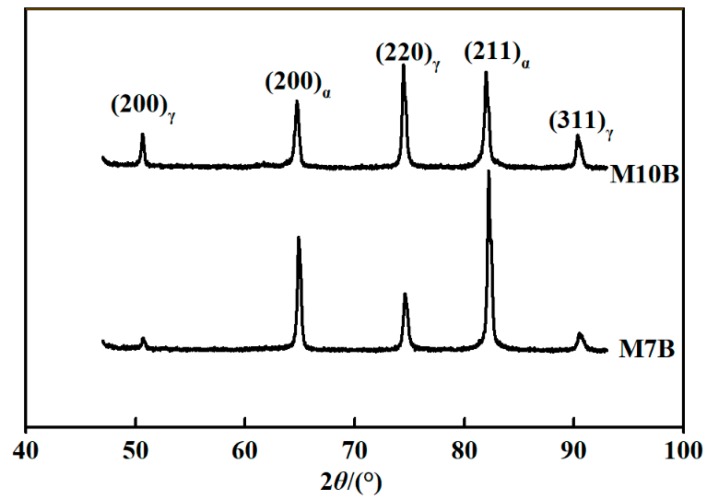
X-ray diffraction (XRD) patterns of M7B and M10B.

**Figure 2 materials-13-01304-f002:**
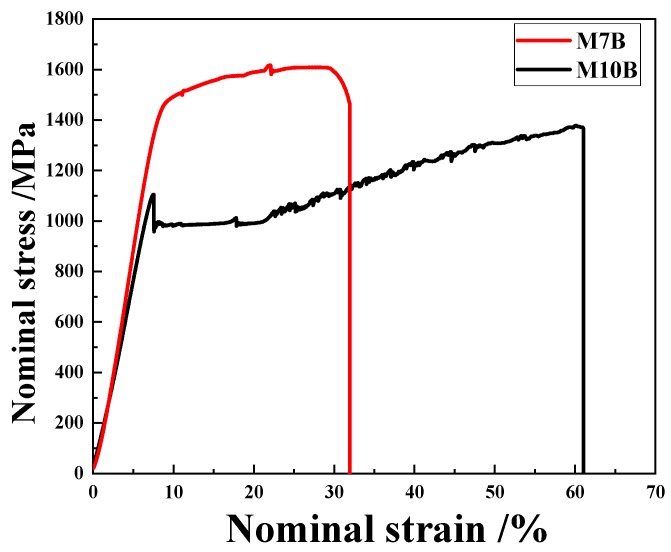
Engineering stress–strain curves of M7B and M10B.

**Figure 3 materials-13-01304-f003:**
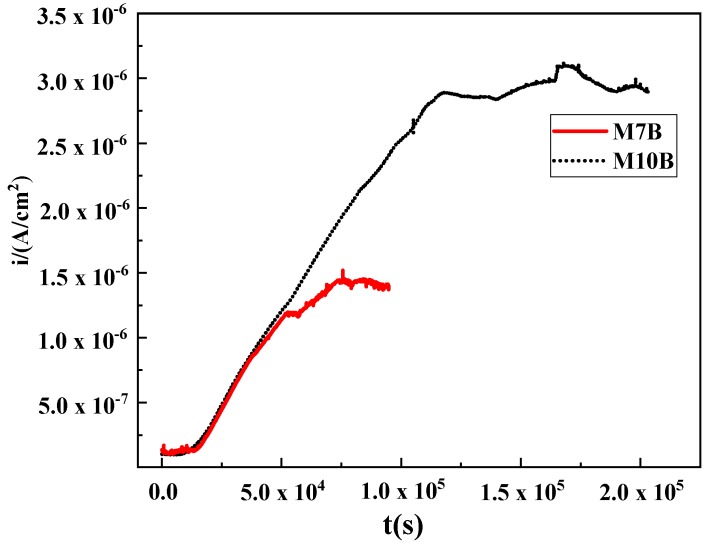
Hydrogen permeation curve of M7B and M10B.

**Figure 4 materials-13-01304-f004:**
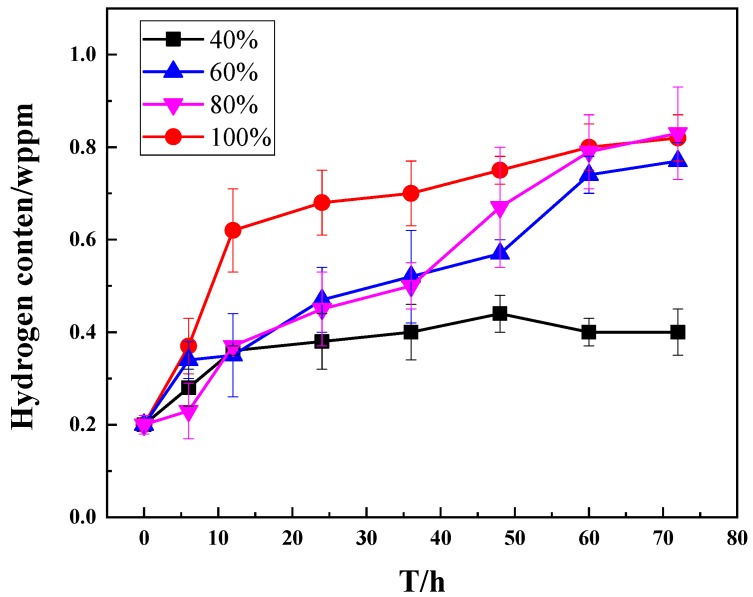
Hydrogen concentration over time after exposure to different humidity conditions for M7B.

**Figure 5 materials-13-01304-f005:**
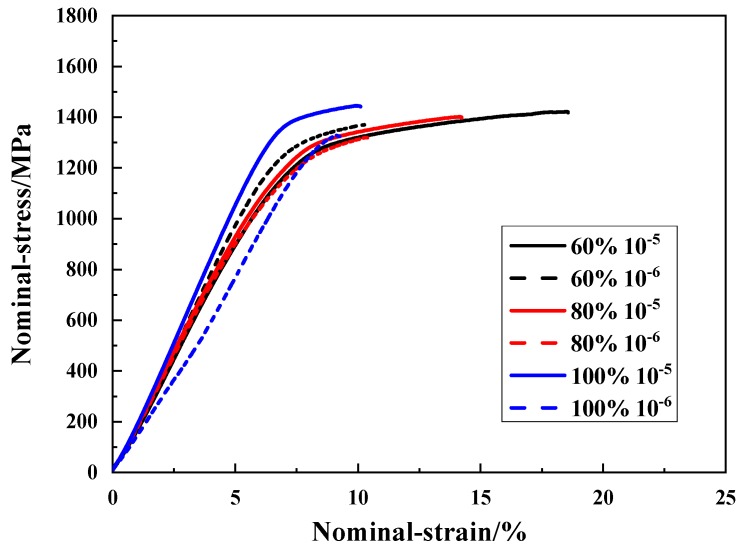
Nominal stress–strain curves under humidity for M7B.

**Figure 6 materials-13-01304-f006:**
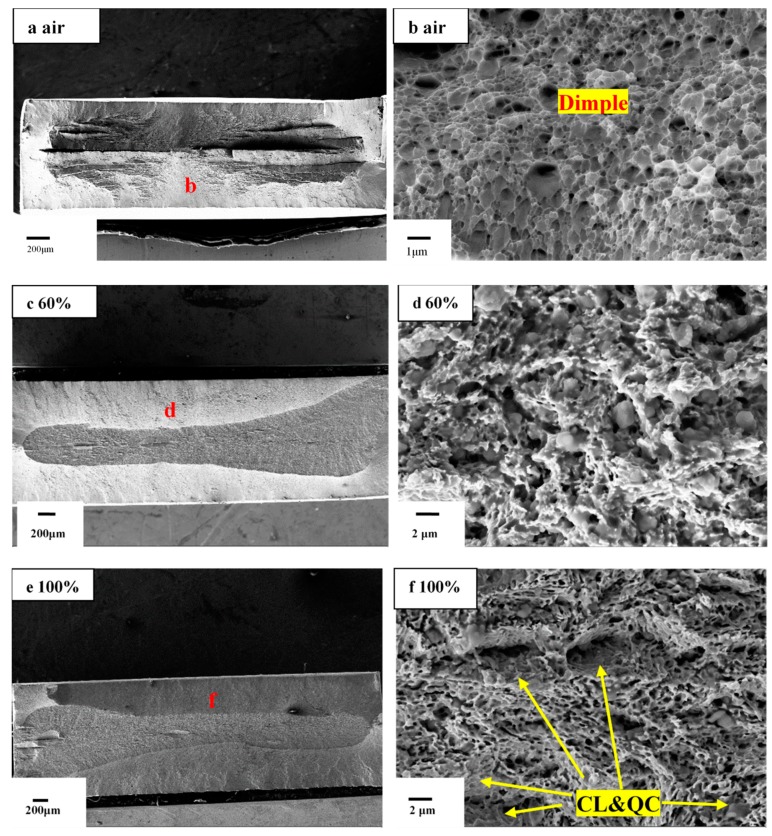
Fracture surface of M7B after SSRTT under different relative humidity (RH) conditions with different strain rates: (**a**,**b**) in air at 10^−4^ s^−1^; (**c**,**d**) in 60% RH at 10^−5^ s^−1^; (**e**,**f**) in 100% RH at 10^−5^ s^−1^; (**g**–**j**) in 60% RH at 10^−6^ s^−1^; (**k**,**l**) in 100% RH at 10^−6^ s^−1^.

**Figure 7 materials-13-01304-f007:**
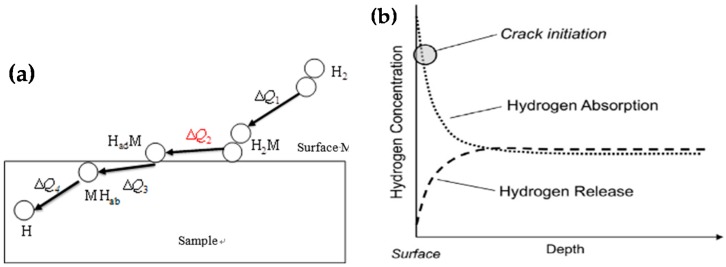
Schematic diagram for hydrogen entry from air to sample (**a**) and the change in the in-depth profile of hydrogen concentration in high strength steel under atmospheric corrosion (**b**) [41].

**Figure 8 materials-13-01304-f008:**
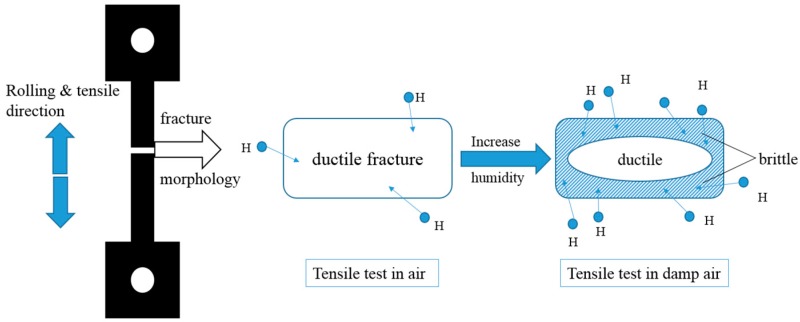
Schematic diagram of the hydrogen influence on the sample fracture.

**Figure 9 materials-13-01304-f009:**
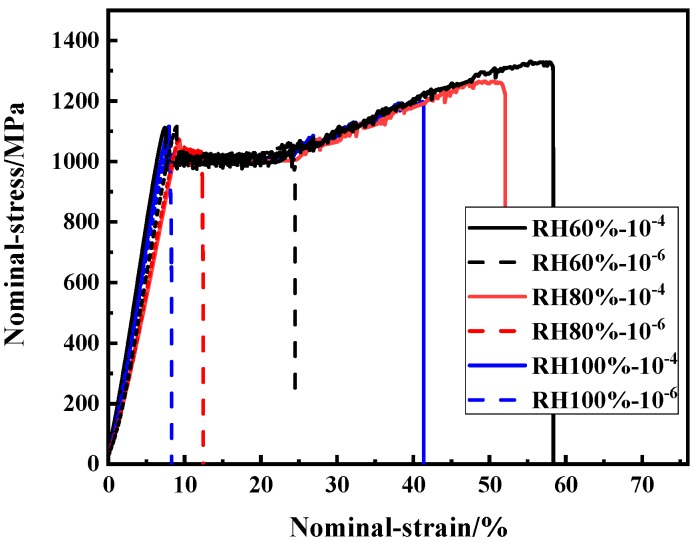
Nominal stress–strain curves under humidity for M10B.

**Figure 10 materials-13-01304-f010:**
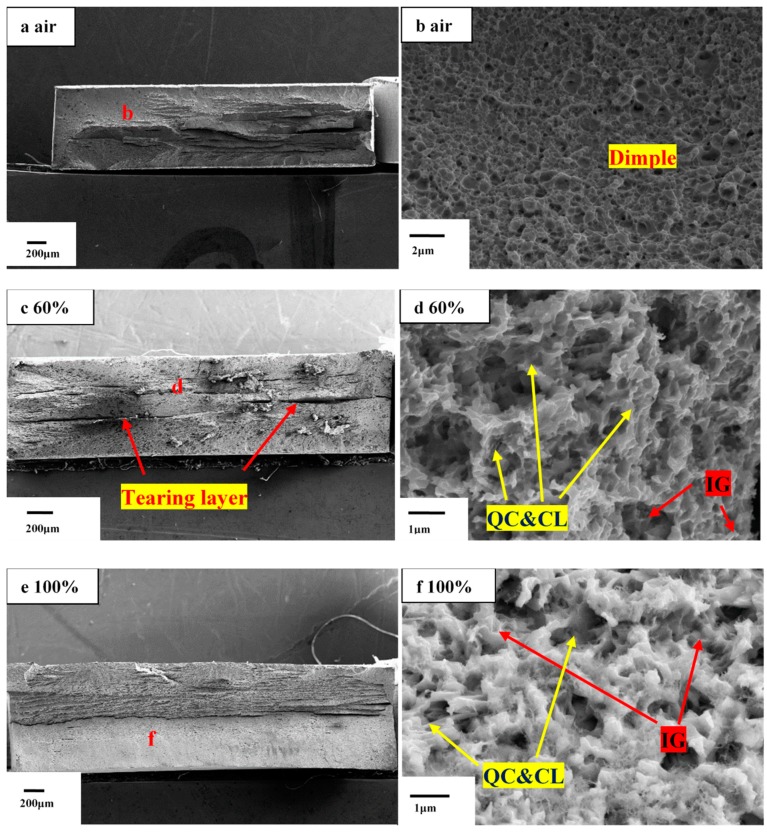
Fracture surface of M10B after SSRTT under different RH conditions with different strain rates: (**a**,**b**) in air at 10^−4^ s^−1^; (**c**,**d**) in 60% RH at 10^−4^ s^−1^; (**e**,**f**) in 100% RH 10^−4^ s^−1^; (**g**,**h**) in 60% RH at 10^−6^ s^−1^; (**i**,**j**) in 100% RH at 10^−6^ s^−1^.

**Figure 11 materials-13-01304-f011:**
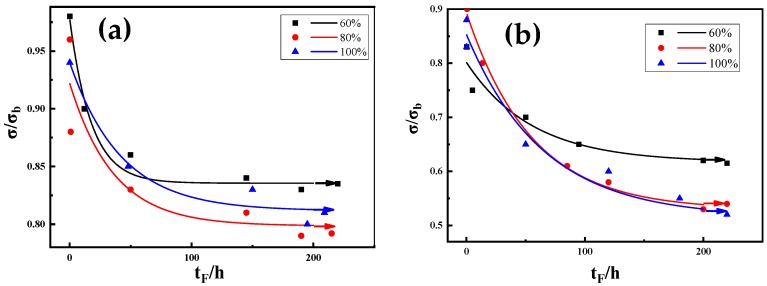
The relationship between applied stress σ/σb and fracture time t_F_, (**a**): M7B and (**b**): M10B.

**Table 1 materials-13-01304-t001:** Chemical composition in wt. % and mechanical properties of medium Mn steel (MMS).

Material/Element	C	Si	Mn	S	Al	V	Fe	Heat Treatment	YS/TS, MPa	Elongation, %	PSE, GPa•%
M7B	0.20	0.24	7.51	0.014	0.035	0.11	Bal	batch annealing 12 h	1450/1614	22.0	35.5
M10B	0.15	0.17	10.4	0.013	1.49	<0.01	Bal	batch annealing 12 h	980/1390	30.3	42.1

**YS/TS:** Yield Strength/Tensile Strength, **PSE:** The product of strength and elongation.

**Table 2 materials-13-01304-t002:** Mechanical properties and sensitivity of hydrogen embrittlement after slow strain rate tensile test (SSRTT) for M7B.

Strain Rate	Humidity	Fracture Strength/MPa	Elongation /%	PSE/ GPa•%	Sensitivity of HE /%	C_H_/wppm
10^−4^ s^−1^	Air (about 40% RH)	1594	22	35.07	0	0.35 ± 0.03
10^−5^ s^−1^	60% RH	1381	15	20.7	31.8	1.02 ± 0.08
	80% RH	1395	13.5	18.8	38.7	1.63 ± 0.05
	100% RH	1455	7.5	10.9	65.9	1.81 ± 0.12
10^−6^ s^−1^	60% RH	1406	9	12.7	59.1	1.68 ± 0.04
	80% RH	1380	8	11.0	63.6	1.92 ± 0.03
	100% RH	1415	6	8.5	72.7	2.32 ± 0.15

**PSE:** The product of strength and elongation, **HE**: hydrogen embrittlement, **C_H_**: hydrogen concentration.

**Table 3 materials-13-01304-t003:** Mechanical properties and sensitivity of hydrogen embrittlement after SSRTT for M10B.

Strain Rate	Humidity	Fracture Strength/MPa	Elongation /%	PSE/GPa•%	Sensitivity of HE /%	C_H_/wppm
10^−4^ s^−1^	Air (about 40% RH)	1380	30.3	41.81	0	0.43 ± 0.04
10^−4^ s^−1^	60% RH	1330	29.2	38.84	3.6	0.48 ± 0.05
	80% RH	1260	22.5	28.35	25.7	0.91 ± 0.14
	100% RH	1200	20.87	25.04	31.1	1.35 ± 0.08
10^−6^ s^−1^	60% RH	1050	10.1	10.61	67.0	2.13 ± 0.21
	80% RH	1030	7.3	7.52	75.9	2.31 ± 0.15
	100% RH	1010	6.6	6.67	78.2	2.23 ± 0.27

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
