# Peer review of "Effect of Relative Humidity on Mechanical Degradation of Medium Mn Steels"

_materials, 2020, doi:10.3390/ma13061304_

Round 1
Reviewer 1 Report
Information stated in the paragraph 3.2. does not correspond with the Figure 4.
How was the sensitivity of hydrogen embrittlement alculated?
Author Response
Response: Thank you for your comments. The authors feel sorry about the mislabeling 60% and 100% in Figure 4, causing the mismatch between statement and the figure. The definition and calculation of hydrogen embrittlement sensitivity was added in section 3.31 (Page 6 Line 197-204).
Reviewer 2 Report
Reviewer’s Report
Manuscript ID: materials-699978
Title: Effect of relative humidity on hydrogen-induced delayed cracking for medium Mn steels
Authors: Qingyang Liu et al.
This paper reports SSRTT and CLT results of two types of medium Mn steels exposed to different relative humidity. A substantial degradation of medium Mn steels is worthy to be discussed, but the quality of the manuscript is poor as an article concerned with the subject.
The title should properly express the content of the paper. The present title focuses on delayed cracking, but SSRT and CLT are equally conducted. Specific discussion about “delayed cracking” is hardly found in the text. Only the amount of austenite is presented as the microstructural difference between M7B and M10B. Additional data on microstructures that characterize the steels are recommended. Mechanical tests are conducted after exposure of specimens for 72h to humid environments. (Please clearly describe whether SSRTT and CLT after the exposure are conducted still under humid environments or not.) According to Fig. 3, 72h (= 2.6×105 sec) is quite before saturation for M10B and the hydrogen distribution in M10B is not likely uniform. The employed experimental condition may not be proper for comparing M7B and M10B. Figure 4 is only for M7B and hydrogen entry is not shown for M10B. Table 1. Adjust the position of the “Elongation” column. Table 2. Show the definition of “sensitivity of hydrogen embrittlement”. Figure 11 totally visualizes Tables 4 and 5. The tables are not necessary. Tensile fracture surfaces are shown in Figs. 6 and 10 with two magnification scales. The authors state the appearance of the brittle edge zone in Fig. 6 as the hydrogen effect, but a common idea is that hydrogen causes earlier fracture in the central part of the specimen. In fact, Figure 9 shows that hydrogen degradation takes place during plastic deformation. The magnification of Fig. 6b is almost twice of Fig. 6d, f, g, h, indicating large dimple size in air, and Fig. 6l is substantially different from others under lower humidity.In order to support the idea of the authors, it is necessary to show details of “brittle fracture surface” with high magnifications. It is also necessary to show locations of the high magnification sites in low magnification views.
The statements on fracture surfaces of M10B, lines 241 - 246, are not clear. Indicate corresponding parts in figures. The meaning of “a combination of toughness and brittleness” cannot be understood.
Figure 7 and related descriptions in lines 202-209 are primitive as a discussion of experimental results. The sentence “according to the mechanisms •• hydrogen embrittlement in steels” is incomplete. Figure 4 shows almost same hydrogen concentrations in M7B irrespective of relative humidity after exposure of 72h. On the other hand, Table 2 shows a substantial dependence of hydrogen embrittlement on relative humidity. The results imply that hydrogen concentration is not a decisive factor for embrittlement. Some explanations are requested. Line 59. Check Yikai et al. [19] Line 98. The meaning of “through constructing a closed space” cannot be understood. Line 221. “Increases from” → “decreases from”.Author Response
Response: Thanks for your pertinent comments. The authors carefully made efforts to address the comments. The specific discussion about the ‘delayed cracking’ was added in the introduction session (Page 2 Line 57-65). The microstructure observation of M7B was discussed in paragraph 2.1 Page 2 Line 83-85 (Liu et al. 2020). The authors feel sorry about the confusion we made and we clarified that SSRTT and CLT tests are conducted still under humid environment in Page 4 Line 143. Figure 3 shows the result of hydrogen permeation test, which is performed to obtain the effective hydrogen diffusion coefficient by hydrogen charging process. As the reviewer said, 72h is quite before saturation for M7B and M10B, that’s the steady-state current density. According to the Ref (Jebaraj, Morrison and Suni 2014) and (Turnbull, Saenz de Santa Maria and Thomas 1989), the steady-state current density is what we need to calculate the effective hydrogen diffusion coefficient by ‘Time-lag method’ (Turnbull, Saenz de Santa Maria and Thomas 1989). There is no need to have hydrogen saturated for samples, hydrogen diffusion is controlled by the concentration gradient (Knapton 1977). While this hydrogen permeation method is performed to obtain the effective hydrogen diffusion coefficient of M7B and M10B. The authors modified the name as “effective hydrogen diffusion coefficient”. Table 2 has been adjusted and the definition of “hydrogen embrittlement sensitivity” has been added in section 3.31 Page 6 Line 197-204. Figure 4 is only for M7B to see the behavior of hydrogen entry and the hydrogen content under different relative humidity. The authors agree with the common idea that hydrogen causes earlier fracture in the central part of the hydrogen pre-charged specimen. But the fracture surface of M7B in this study shows the brittle edge zone in Figure 6, and the only reason is the presence and enrichment of hydrogen at the edge zone of the specimen. The magnification of Fig. 6b is almost twice of Fig. 6d, f, h, l, indicating small dimple size in air.
Thank you for your kind remind. To make the statement more clear, the arrows and captions are added in Figure 6 and 10. The statements on fracture surfaces of M10B has been modified in Page 11 Line 270-278. The statement of ‘a combination of toughness and brittleness’ has been deleted and restated.
Description related to Figure 7 have been added in Page 10 Line 235-242. The authors agree that Figure 4 shows almost same hydrogen concentrations in M7B irrespective of relative humidity after exposure of 72h. We added one column of hydrogen concentration in Table 2 and 3, the hydrogen content of specimen after SSRTT varies from that of exposure test. The result shows the hydrogen embrittlement and hydrogen content increase with the increase of relative humidity. As the tensile tests went on and stress loaded, more and more hydrogen entered the specimen and resulted in hydrogen degradation. And the corresponding explanations are added in 3.31 and 3.32.
The statement about corrosion dynamic model has been corrected as ‘A corrosion kinetic model was also discussed by Yikun in relation to atmospheric corrosion in dynamic environments and the result demonstrated that the relative humidity was the most influential factors on corrosion’ in Line 69-70. The meaning of “through constructing a closed space” has been modified to “for enclosed space" in Page 3 Line 112.
Other comments have also modified in the manuscript.
Reviewer 3 Report
While the manuscript deals with an interesting issue related to assess the mechanical properties of medium manganese steel considering relationship between sensitivity of hydrogen embrittlement and relative humidity, there are a few questions that need to be clarified before further consideration for a publication. In addition, authors should improve the grammar and style.
Authors describe that “The tensile rate was 10-4 s-1 in air, 10-5 s-1 for M7B and 10-118 6s-1 for M10B in wet air.” The results of tensile test are influenced by the tensile rate. Therefore, the quantitative comparisons for various humidity require mechanical properties at the same tensile rate condition. To consider relative humidity, authors should compare with mechanical properties at the same tensile rate (Tables 2 and 3).
Authors explain that “At 100% RH, hydrogen concentration changes with the fastest growth rate and reaches saturation in a relatively fast time.” However, the hydrogen concentration changes of 60% RH are faster than that of 100% RH in Figure 4. Authors should clearly explain the Figure 4.
Authors describe that “When M7B steel is in service under humidity conditions, the macroscopic fracture reveals a middle arc shape similar to the "sole" shape, shown in Fig 6(c)(e)(g)(k), in which the edge is a brittle fracture zone with cleavage characteristics, while the center is a ductile fracture caused by rapid cracking.” However, authors did not explain the Figure 6 (d), (f), (h), (i), (j) and (l). In case of Figure (b), (d), (f), (h), (i), (j) and (l), authors did not indicate the location of measurement. Authors should explain more about Figure 6.
Authors describe that “According to the experimental results, with the increase in RH, the fracture strength of M10B steel changes significantly.” However, there is no difference for fracture strength in slow strain rate. Authors should add the explanation.
Authors write that “It can be seen that the increase in RH and 222 the decrease in strain rate will lead to an increase in the hydrogen embrittlement sensitivity of the 223 material, thus reducing its service performance.” However, this sentence is duplicated in Line 230 ~ 232. Authors should modify the sentence.
Please consider the few minor comments:
In manuscript, authors should replace Mpa with MPa. In section 2.3, authors should add the information of tensile test specimen. In Figure 4, authors should define the y-axis label. In Figure 2, 5 and 9, authors should unify the x and y axis labels. In Figure 10 (b), (d), (f), (h) and (j), authors should unify the scale. In addition, authors should remark the location of measurement. In Table 5, authors should replace for. with for.Author Response
Response: Thank you for your contribution on this study.
Firstly, there should be some mistakes describing the tensile rate of two steels. The statement has been modified ‘The tensile rate was 10-4 s-1 in air, 10-5 s-1 and 10-6s-1 for M7B in wet air, 10-4 s-1and 10-6s-1 for M10B in wet air’ in Page 4 Line 135-137.
The authors feel sorry about the mislabeling 60% and 100% in Figure 4, so the statement of Figure 4 is correct.
Thank you for your kind remind, we have added explanations, arrows and captions on Figure 6 (Line 232-243) and Figure 10 (Line 271-279) in the manuscript. That’s true that there is no difference for fracture strength of M10B in slow strain rate, modified explanation has been added in section 3.32.
Thank you for your kind remind, the repeated statement in Line 222-224 has been modified to ‘The increase of RH and decrease of strain rate accelerate the hydrogen degradation of material by reducing the elongation and increasing the hydrogen embrittlement sensitivity’ in Line 259-261.
Also, the authors appreciate the minor comments you made. We corrected the clerical error of the ‘MPa’, added the information of tensile test specimen in section 2.3 Line 140-141. The definition of y-axis label of Figure 4 is diffusible hydrogen content, which can be found in section 2.2 and 3.2. The x and y axis labels in Figure 2, 5 and 9 have been unified. Other modifications are also made to address the comments.
Reviewer 4 Report
The manuscript under evaluation analyzes the effect of relative humidity on hydrogen-induced delayed cracking for medium Mn steels. Two different alloys are tested at diverse conditions, discussing the results generated after characterization by multiple techniques. In general, the manuscript is certainly interesting for potential readers and discussion is well-supported by experimental results. In this manner, I suggest the publication of the work. However, some aspects need to be corrected after a minor revision before to be definitely accepted. The following items should be addressed:
Introduction section is too short. Please, expand this section explaining better the interest of the study, available previous works in literature and applications of findings. In experimental section, please, indicate in a clearer way conditions of the as received materials and treatments carried out by you. Some doubts can be arisen during the lecture (mainly in paragraph between lines 70-80). Tables 1 and 2. Use acronyms in titles of two last columns and define them in the body text. Conclusions should be revised and re-written. Most of the sentences are not gramatically correct. Please, consider use help for a native English speaker. Revise carefully the entire manuscript in order to find some English language and style fine/minor mistakes. One more time, help of a native English speaker would be suggested.Author Response
Response: Thank you for your valuable comments and your approval and recognition of our work. The manuscript has been modified by addressing your comments. We have expanded the introduction section and discussed more about the hydrogen-induced delayed cracking phenomenon in Page 2 Line 57-65. We added explanation about the as received materials and treatment in manuscript. The material was received in annealing state and we did not do any treatment after.
Thank you for your suggestion, acronym is using in Table 1 and 2 to make the Tables look better. The conclusion part has been carefully revised and re-written with the help of a native English speaker.
Reviewer 5 Report
Dear authors,
a very nice study on the hydrogen induced cracking of medium mn steels.
I have just one comment.
Can you please make a comment in the paper on the number of specimen used for each experiment and please adress the scatter in the experimental results.
Regards,
a Reviewer
Author Response
Response: The authors appreciate the valuable comment and added the explanation in the manuscript. Again thanks for your approval and recognition of our work. The statement ‘Due to the dispersion of experimental data for the hydrogen test, the average value of three samples was used and scale bar was shown in Figure 4’ was added in section 2.2. Two parallel samples are used in CLT and SSRTT and the results showed great consistency.
Round 2
Reviewer 2 Report
- Technical errors likely on submission. Same four Fig.6 and six Fig. 10 are present.
- Insufficient information. Describe details (location, size, preparation) of the specimens prepared for hydrogen concentrations in Tables 2 and 3.
- Insufficient information. Describe the thickness of specimens used for Fig. 3 and 4.
- Uncertain literature. The name of Yikun is not found in Ref. [22]. Check p. 2 line 69.
- Title. In this paper CLT as compared with SSRT is just a part of the work. Focusing on delayed cracking is not found. Change of the title to, for example, “Effect of relative humidity on mechanical degradation of medium Mn steels”, is suggested.
- P.2 line 57, “While another ••• the presence of hydrogen”. Incomplete sentence
- P. 5 line 169. Be careful that Eq. (1) is for the case of a uniform solution without trapping according to the McNabb-Foster equation. A. McNabb and P. K. Foster: Trans. Metall. Soc. AIME, 227, 618(1963).
- P. 6 line 195 “the fracture strength of M7B steel does not change significantly” and line 208 “••• strain rate have little effect on the fracture strength”. Physically meaningful is the true fracture stress. The true stress increases with elongation associated with the reduction of area even though the nominal stress is the same.
- P.13 line 249. Describe precisely the features that the authors judged as QC and IG. Fig. 6f looks like fine dimples, and a difference between QC and IG indicated in Figs. j and l is difficult to find.
- P.13 line 251 and Fig. 7. For most readers, Fig. 7 and the description from “Due to the increase in RH •••” to the end of line 256 are too elementary and may not be necessary. In this study, crucial is the hydrogen diffusion depth and concentration.
- P.13 line 259. Major mechanisms of hydrogen embrittlement so far proposed are HEDE, HELP and HESIV (hydrogen-enhanced strain-induced vacancies, M. Nagumo and K. Takai: The predominant role of strain-induced vacancies in hydrogen-embrittlement of steels: Overview, Acta Mater., 165(2019), 722-733.) The mechanism is beyond the scope of the present study and no result is found to support or refute a specific mechanism. The sentence “And the influence ••• hydrogen-enhanced decohesion [34] in steels.” gives no information about this study and to be deleted.
- P.15 line 283-285. Correct the disordered layout of sentences “By comparing the tensile facture ••• ; see” and following. The logic of the sentence that “the SEM analysis for M10B was carried out by comparing the tensile fracture of M7B” cannot be understood.
- P. 19 line 306. Delamination may originate in the fabrication process of the plate. Ref. [35] is not available but the situation there may be quite different from the present case. (Check the journal name of Ref. [35])
- P.19 line 312. Describe precisely the features that the authors judged as QC and IG in Fig. 10d. Also for IG in Figs. 10f, 10h, 10j.
- P. 23 line 339 and Tables 4 and 5. Information in the Tables duplicates that in Fig. 11. The tables are superfluous.
- P 24 line 368. The TRIP effect is feasible, but no experimental evidence is presented in this paper.
Author Response
Thank you for your effort to our study. Comments to our manuscript (Ref: materials-699978) are very helpful and constructive. We have carefully revised the manuscript according to the comments. The changes in the revised manuscript were highlighted with yellow background. Followings are our point-to-point responses with green background to the comments.
1. Technical errors likely on submission. Same four Fig.6 and six Fig. 10 are present.
Response: The authors do apologize for the mistakes that we made in the manuscript and thank you for pointing them out. We have modified the errors.
2. Insufficient information. Describe details (location, size, preparation) of the specimens prepared for hydrogen concentrations in Tables 2 and 3.
Response: Details of the specimens for hydrogen concentration was added in Page 4 Line 138-141.
3. Insufficient information. Describe the thickness of specimens used for Fig. 3 and 4.
Response: The author described the thickness of specimens for Fig.3 and 4 in Page 3 Line 102 and Line 117, respectively.
4. Uncertain literature. The name of Yikun is not found in Ref. [22]. Check p. 2 line 69.
Response: the name of Yikun has been replaced by ‘Cai’ in Page 2 Line 69.
5. Title. In this paper CLT as compared with SSRT is just a part of the work. Focusing on delayed cracking is not found. Change of the title to, for example, “Effect of relative humidity on mechanical degradation of medium Mn steels”, is suggested.
Response: Thank you for your suggestion, the authors also think the suggested title is more appropriate.
6. P.2 line 57, “While another phenomenon which is a common problem in this material is strength degradation due to the presence of hydrogen”. Incomplete sentence
Response: Sorry for the confusion that it caused. The complete sentence is ‘Another phenomenon for this type of material is the strength decline due to the presence of hydrogen’, shown in Page 2 Line 57.
7. P. 5 line 169. Be careful that Eq. (1) is for the case of a uniform solution without trapping according to the McNabb-Foster equation. A. McNabb and P. K. Foster: Trans. Metall. Soc. AIME, 227, 618(1963).
Response: Thank you for your comment. The authors think that it is necessary to add the mentioned reference in the manuscript, in Page 5 Line 172-173.
8. P. 6 line 195 “the fracture strength of M7B steel does not change significantly with an increase in RH, but the elongation decreases accordingly’” and line 208 The change of RH and strain rate have little effect on the fracture strength of M7B, but significant effect on the elongation and hydrogen embrittlement sensitivity”. Physically meaningful is the true fracture stress. The true stress increases with elongation associated with the reduction of area even though the nominal stress is the same.
Response: Thank you for your kind reminding. It is absolutely true that the true stress changes due to the reduction of the area. For the tested samples under various RH, especially with the strain rate of 10-6s-1, the elongation rates were very close. Therefore, the reduction of the sample area did not affect the final fracture strength that much. Nevertheless, we do agree with the reviewer that true stress changes during the tensile test.
9. P.13 line 249. Describe precisely the features that the authors judged as QC and IG. Fig. 6f looks like fine dimples, and a difference between QC and IG indicated in Figs. j and l is difficult to find.
Response: For Fig. 6f, j and l, they actuarially showed mixed modes of QC or IG with dimples. There were areas which show QC or IG with dimples around. With the images in the PDF file, it may not be seen that clearly, but QC or IG characteristics can be identified. More description about the features to judge as QC and IG are added in Page 9 Line 240-248.
10. P.13 line 251 and Fig. 7. For most readers, Fig. 7 and the description from “Due to the increase in RH •••” to the end of line 256 are too elementary and may not be necessary. In this study, crucial is the hydrogen diffusion depth and concentration.
Response: Thank you for your revision and comments. The authors rewrote the statement in Page 10 Line 249-250 and tried to make sense of it. We also added some statements about the hydrogen concentration at the tip of crack in Page 10 Line 258.
11. P.13 line 259. Major mechanisms of hydrogen embrittlement so far proposed are HEDE, HELP and HESIV (hydrogen-enhanced strain-induced vacancies, M. Nagumo and K. Takai: The predominant role of strain-induced vacancies in hydrogen-embrittlement of steels: Overview, Acta Mater., 165(2019), 722-733.) The mechanism is beyond the scope of the present study and no result is found to support or refute a specific mechanism. The sentence “And the influence ••• hydrogen-enhanced decohesion [34] in steels.” gives no information about this study and to be deleted.
Response: Thank you for your comments, the authors totally agree with you about the major mechanisms of hydrogen embrittlement. And the authors decided to delete the sentence which is not relevant to the study.
12. P.15 line 283-285. Correct the disordered layout of sentences “By comparing the tensile facture ••• ; see” and following. The logic of the sentence that “the SEM analysis for M10B was carried out by comparing the tensile fracture of M7B” cannot be understood.
Response: The authors do apologize for the layout errors and we have modified the layout. And the clerical error of ‘M7B’ has been replaced by ‘M10B’ in Page 11 Line 279.
13. P. 19 line 306. Delamination may originate in the fabrication process of the plate. Ref. [35] is not available but the situation there may be quite different from the present case. (Check the journal name of Ref. [35])
Response: Sorry for the wrong format of this reference and the authors checked the journal name and modified it as Ref [40] in manuscript Page 11 Line 286-288 (Tao, J.et.al. 2016. A study of the mechanism of delamination fracture in bainitic magnetic yoke steel. Materials & Design 108, 429-439.). The reason of the delamination fracture is that stress concentration in thickness direction at the middle of thickness accelerates the occurrence of the crack source in the half thickness of the cross section of the specimen during tension test.
14. P.19 line 312. Describe precisely the features that the authors judged as QC and IG in Fig. 10d. Also for IG in Figs. 10f, 10h, 10j.
Response: The features of QC and IG are added in Page 9 Line 240-248. Like the authors mentioned previously, the fracture morphology was a mixture of various modes. The characteristics of IG is obvious in Fig 10 (f) (h) (j) with clear trail of grain fall-off.
15. P. 23 line 339 and Tables 4 and 5. Information in the Tables duplicates that in Fig. 11. The tables are superfluous.
Response: Thank you for your reminding. The authors deleted the superfluous tables in the manuscript and added comments in Page 13 Line 305.
16. P 24 line 368. The TRIP effect is feasible, but no experimental evidence is presented in this paper.
Response: Thank you for your comments. The original statement has been modified to “The elongation of M10B is higher than M7B, because more austenite tends to deform during the tensile test” Page 14 Line 331.
